# Choline-Mediated Regulation of Follicular Growth: Interplay Between Steroid Synthesis, Epigenetics, and Oocyte Development

**DOI:** 10.3390/biology14091220

**Published:** 2025-09-08

**Authors:** Wenfeng Liu, Xucheng Zheng, Haiming Yang, Zhiyue Wang

**Affiliations:** College of Animal Science and Technology, Yangzhou University, Yangzhou 225009, China; dx120220169@stu.yzu.edu.cn (W.L.); dx120230175@stu.yzu.edu.cn (X.Z.); hmyang@yzu.edu.cn (H.Y.)

**Keywords:** choline, follicle development, DNA methylation, steroid hormones

## Abstract

This review delves into the impact of choline, a vital feed additive, on follicular development and reproductive performance. By blending animal nutrition, epigenetics, and reproductive endocrinology, this study uncovers how choline regulates follicular hierarchy and maturation. Adequate dietary choline is shown to enhance follicle maturation and boost reproductive success. Conversely, both insufficient and excessive choline intake can negatively affect follicular development, leading to issues like increased follicle atresia. Additionally, the research highlights the importance of establishing optimal choline supply systems. It suggests that future studies should focus on determining precise choline requirements and understanding the related metabolic and epigenetic mechanisms. Overall, this work provides valuable insights into improving animal reproduction through better choline nutrition.

## 1. Introduction

Reproductive dysfunction poses a common challenge in both clinical medicine and animal husbandry, with follicular health and developmental quality being key determinants [1,2]. Follicular development not only affects oocyte formation but also directly influences fertilization rates and embryonic survival, thereby determining overall reproductive efficiency [3]. As a crucial component of cell membranes and a key player in one-carbon metabolism, choline is essential for normal follicular development. Studies indicate choline deficiency may impair membrane synthesis and elevate homocysteine (HCY) levels, consequently disrupting follicular growth and function [4,5]. In addition, the role of long non-coding RNAs (lncRNAs) in reproductive health has gained attention in recent studies, with LncRNA Gm2044 identified as a key regulator in germ cell development. Choline’s involvement in regulating such lncRNAs could represent a critical pathway influencing folliculogenesis [6]. Moreover, choline participates in DNA methylation and gene expression regulation—indispensable biological processes during folliculogenesis [7,8]. Recent advancements have highlighted the importance of epigenetic modifications in regulating reproductive health, and choline’s role in these processes, particularly through its involvement in super-enhancers and transcription factor regulation, is gaining attention [9].

Although existing research has explored choline’s general effects on animal growth and health, its specific roles in follicular development remain insufficiently investigated. As an essential nutrient, choline’s involvement in the regulation of epigenetic mechanisms and homocysteine metabolism has been observed in several animal models, suggesting its broader influence on folliculogenesis [10]. Additionally, environmental stressors, including heat stress, have been shown to negatively affect choline metabolism and, consequently, follicular development. Understanding the intricate relationship between choline and reproductive health, especially under environmental stress, remains an area that warrants further research [10,11].

This review systematically examines choline’s functions in folliculogenesis, analyzing how it influences follicular development through one-carbon metabolism, homocysteine regulation, and epigenetic modulation pathways.

## 2. Literature Review Methodology

This manuscript is a mini-review aimed at summarizing the effects of choline on follicular development. The literature search and selection followed strict criteria to ensure comprehensive coverage of relevant studies.

### 2.1. Literature Search Timeframe and Databases

The literature search covered studies from 1971 to 2025, using several databases for retrieval, including PubMed, Web of Science, and Google Scholar. These databases are widely used in the life sciences and medical fields, ensuring the breadth and reliability of the literature.

### 2.2. Search Keywords

The following keywords were used in the search: “choline”, “follicular development”, “DNA methylation”, “epigenetics”, “reproductive performance”, “follicle maturation”, etc. These keywords were combined to select the most recent studies relevant to choline and follicular development.

### 2.3. Literature Selection and Evaluation Criteria

The selection criteria included experimental studies that explored the direct or indirect effects of choline on follicular development, research involving various species (e.g., poultry, mammals), and studies employing in vitro or animal models. Review articles and irrelevant studies were excluded to maintain a focus on primary experimental data and observations.

### 2.4. Literature Assessment

All included articles were independently reviewed by two researchers, who assessed the quality and relevance of each article. Any doubts were discussed further to ensure the integrity of the selected literature.

## 3. Choline Structure, Absorption, and Metabolism

Choline (trimethyl-β-hydroxyethylammonium) is a positively charged quaternary ammonium compound that serves as a critical one-carbon metabolite in cellular methylation, acetylcholine synthesis, and phosphatidylcholine (PC) biosynthesis [12,13], consisting of a nitrogen atom bonded to four alkyl or other groups with excellent water solubility.

Dietary choline primarily exists as PC > 90%, with <10% present as free choline or sphingomyelin [14]. Choline absorption mainly occurs in the jejunum and ileum, facilitated by specialized transporters [15]. Dietary conjugated choline forms free choline through deacetylation catalyzed by pancreatic lipase and mucosal lipase with assistance from intestinal epithelial transporters [16]. Quantitatively, pancreatic lipase dominates choline digestion [17]. However, approximately 50% of dietary PC escapes intestinal enzymatic digestion, entering lymphatic vessels intact. The digestive process converts most ingested PC into lysophosphatidylcholine (LPC) for absorption. Within intestinal wall cells, LPC can either deacylate into glycerophosphocholine or re-acylate into PC [18]. Choline absorption occurs in the jejunum and ileum via energy- and sodium-dependent transporters in the intestinal epithelium [19]. In guinea pigs, ileal cells absorb choline approximately three times faster than jejunal cells [20], as PC primarily hydrolyzes into choline in proximal and middle small intestine segments [21]. Choline absorption does not compete with other nutrients; only one-third of dietary choline is absorbed intact, while the remaining two-thirds undergo microbial metabolism into betaine and trimethylamine (TMA) [22,23]. Additionally, choline can cross membranes via passive diffusion. Absorption efficiency depends on individual nutritional status, dietary choline intake, and the presence of other intestinal substances like fats [24]. After absorption, choline transports into lymphatic circulation as PC–chylomicron complexes and then distributes to tissues primarily as phospholipids bound to plasma lipoproteins [25,26].

Choline plays a critical role in membrane synthesis and methylation pathways. It is primarily involved in the production of phosphatidylcholine (PC) and phosphatidylethanolamine (PE), crucial for cellular function [27,28,29,30,31,32]. Being the primary source of membrane phospholipids, newly absorbed choline rapidly converts into PC via the cytidine diphosphate (CDP)–choline pathway catalyzed by choline kinase α [33]. All cells express at least one of three protein families for choline transport, except erythrocytes and the blood–brain barrier, which use facilitated diffusion [34,35], high-affinity choline transporter 1, medium-affinity choline transporter-like proteins (CTL), and low-affinity polyspecific organic cation transporters [36,37]. Once formed, betaine cannot revert to choline. After donating one methyl group to HCY, betaine generates methionine (MET) and dimethylglycine (DMG) [38,39]. DMG loses one methyl group, forming sarcosine, and then another, forming glycine. A minor choline fraction acetylates into acetylcholine for neural signaling and muscle control [40], catalyzed by choline acetyltransferase, the activity of which depends on choline availability and acetyl-CoA levels [41,42,43]. Choline is metabolized by intestinal microbiota into trimethylamine (TMA), which is further oxidized into trimethylamine N-oxide (TMAO), a compound associated with cardiovascular risk [44,45]. These studies demonstrate choline’s complex metabolic roles.

## 4. Impact of Choline on Follicular Development

### 4.1. Deficiency

In non-pregnant women with a daily choline intake of 480 mg, the CDP–choline pathway is prioritized for PC biosynthesis, while the betaine synthesis pathway is systemically suppressed [31]. Adequate PC is vital for cell membrane integrity and cell survival. Choline transport into cells is inhibited by caspase-3-dependent and -independent mechanisms, which lead to an overall suppression of choline incorporation into the CDP–choline pathway [46]. During follicular development, PC accumulation is critical for follicular development, and its depletion can significantly impact follicular development and follicular cavity formation [47]. Both the CDP–choline pathway, which uses choline as a substrate, and the PEMT-PC pathway, which uses SAM as a methyl donor, have high metabolic priority [48,49]. SAM is primarily used for PEMT-PC synthesis. Pregnant and lactating women with low choline intake show increased CDP-PC synthesis of different genotypes and a weakened PEMT-PC pathway, which may lead to methyl donor deficiency and affect the body’s methylation level. Under choline deficiency and the absence of MET, folic acid, vitamin B_6_, and vitamin B_12_, there are significant functional impairments in follicular growth and oocyte maturation [50,51,52]. This can reduce embryo developmental potential and cause stage-specific developmental arrest. These phenomena are attributed to impaired regulation of the methylation process of key maternal imprinting genes (e.g., mesoderm-specific transcript, *MEST*) during folliculogenesis [53,54]. Adding high-dose SAM to culture media causes hypermethylation of genes encoding key DNA methylation factors, including DNA methyltransferases 3 beta (DNMT3B), in cells [55]. Polycystic ovary syndrome (PCOS), a metabolic disorder, can alter serum levels of various metabolites, thereby reducing fertility in patients [56]. PCOS patients exhibit significantly reduced plasma and follicular fluid choline levels, lower choline and phosphocholine in follicular fluid, and abnormal cleavage dynamics [57,58]. This suggests that choline may play a role in oocyte competence development, potentially involving the metabolic process of 1-(5Z, 8Z, 11Z, 14Z, 17Z-eicosapentaenoic)-cn-glycerol-3-phosphocholine [59].

### 4.2. Toxicity

Excessive free choline intake can result in a fishy odor, vomiting, sweating, salivation, hypotension, and liver toxicity. It has been demonstrated that choline consumption can increase the production of TMAO [60,61]. In oocytes, inhibition of β-oxidation can cause insufficient adenosine triphosphate (ATP) synthesis, which in turn can induce cytoplasmic maturation defects and meiotic arrest [62,63]. Therefore, elevated TMAO levels and decreased acetoacetic acid levels can serve as potential indicators of oocyte quality. Studies have shown that in women with repeated implantation failure and PCOS, increased choline levels can promote successful pregnancy, and TMAO levels are negatively correlated with low fertilization rates and poor-quality embryos [64], which is consistent with the findings of a previous study [65]. Research has also indicated that TMAO is present in large amounts in follicular fluid. However, dietary precursors of TMAO, choline, and L-carnitine are not directly related to oocyte and embryo quality. Moreover, high-quality embryos are linked to lower TMAO and γ-butyrobetaine levels in follicular fluid. This suggests that TMAO is associated with poor reproductive outcomes [64]. In summary, these findings indicate that excessive choline metabolites, such as TMAO, can enter the follicular fluid and potentially affect oocyte development.

### 4.3. Optimal Choline Levels for Follicular Health

In Japanese quail (Coturnix japonica), the optimal dietary choline requirement appears to vary depending on the evaluated trait. Evidence indicates that a dietary choline level of approximately 0.126% (1260 mg/kg) can effectively improve egg weight and feed conversion efficiency [66]. In contrast, another experiment demonstrated that 1500 mg/kg of choline is sufficient to maintain laying performance and egg quality, whereas a higher level of 3500 mg/kg is necessary to markedly enhance yolk antioxidant capacity. These findings suggest that the optimal choline level may differ according to the specific performance or quality parameter considered [67]. In laying geese, dietary choline supplementation within the range of 784–913 mg/kg has been shown to enhance reproductive performance, egg production, and sex hormone levels, suggesting this range as optimal for promoting ovarian development and hormone synthesis [68]. For gestating sows, a dietary choline level of approximately 1910 mg/kg has been shown to optimize feed intake, improve birth weight and litter uniformity, and enhance antioxidant capacity and gut microbiota composition [69]. In pigs, dietary supplementation with 500–1000 mg/kg of choline has been shown to improve ovarian function, promote corpus luteum formation, and regulate the expression of reproduction-related genes, thereby enhancing reproductive performance [70]. In transition Holstein cows, supplementation with 60 g/day of choline (ReaShure^®^, Balchem Corporation, New Hampton, NY, USA) was shown to reduce inflammatory gene expression in follicular cells, suggesting that this level supports a healthier follicular environment during early postpartum [71]. In ewes, daily supplementation with 1.60 g/ewe of rumen-protected choline during the periconceptional and late gestation periods effectively reduced embryonic loss and increased lamb birth weight, suggesting this as the most suitable level [72]. In lactating women, an intake of 930 mg/day of choline has been shown to increase the concentration of PEMT-derived choline metabolites in breast milk, supporting enhanced choline delivery to the infant [73]. In pregnant women, prenatal supplementation with 550 mg/day of choline has been shown to enhance plasma concentrations of choline metabolites and promote fetal phospholipid synthesis via the PEMT pathway, making it a recommended intake level to support maternal–fetal health during gestation [74]. Collectively, these findings highlight that choline is not only essential for maintaining basic productivity but also plays pivotal roles in reproduction, growth, and transgenerational health. Therefore, the optimal level should be determined by species, physiological stage, and the targeted trait of interest. A summary of the optimal dietary choline levels across species and their associated reproductive or physiological effects is provided in Table 1.

## 5. Mechanism of Choline in Follicular Development

SAM plays a key role in DNA methylation and is central to reproductive and metabolic regulation [75,76]. Choline, through epigenetic modifications, directly affects key factors in follicular development, such as follicle-stimulating hormone receptor (*FSHR*) and cytochrome P450 family 19 subfamily A member 1 (*CYP19a1*), thereby coordinating oocyte maturation and granulosa cell function [77,78]. SAM also regulates steroidogenic factor 1 (*SF-1*) gene promoter methylation, influencing steroidogenesis in the adrenal–ovarian axis [79]. In PCOS, choline deficiency exacerbates the hypomethylation of genes like *CYP19a1* [80,81], causing hormonal imbalances [82]. Choline supplementation can restore methylation patterns, thereby stabilizing metabolism and endocrine function. Choline and betaine metabolism, linked to folate metabolism, modulates HCY levels and DNMT activity [83,84,85]. This interaction affects reproductive function by regulating the methylation of genes such as *FSHR*, *AR*, and *Cyp11a1* [86,87,88]. Overall, these mechanisms form a choline-mediated metabolic–epigenetic network, suggesting that choline supplementation may enhance follicular development.

### 5.1. Choline and DNA Methylation

As a co-factor for DNMTs and histone methyltransferases, SAM directly regulates the promoter methylation of reproduction-related genes. During folliculogenesis, global DNA methylation reprogramming occurs. Choline intake regulates *DNMT1* expression. In rats, choline deficiency causes a 1.5-fold upregulation of *DNMT1*, leading to global DNA hypermethylation [89]. Moreover, in the livers of E17 fetuses with choline deficiency, the differentially methylated region 2 of the insulin-like growth factor 2 (*Igf2*) gene becomes hypermethylated, and the methylation level is linked to *DNMT1* expression. Choline deficiency may activate a compensatory mechanism that upregulates *DNMT1* to maintain or enhance DNA methylation, affecting specific gene expression. A similar mechanism applies to poultry. From E7 to E18 in chicken embryos, DNA methylation levels change significantly, with hypermethylated regions concentrated in the promoters of development-related genes [90]. In Langshan chickens, whole-genome bisulfite sequencing (WGBS) identified 5948 differentially methylated regions (DMRs) in the hypothalamus and 4593 in the ovary. These DMRs are directly related to reduced fertility, causing a 20% decrease in reproductive performance [91]. In cattle, adding 1.8 mM choline to in vitro cultures increased postnatal body weight and survival rate by 8%, attributed to the hypermethylation of imprinted genes [92]. Similar to poultry, dynamic methylation changes in chicken embryos during organogenesis indicate that choline can alleviate the “reproductive bottleneck” [90,93]. In the Langshan chicken, ovarian DMRs in the hypothalamus and ovary are negatively correlated with fertilization rates [91]. However, most data come from rats or cattle, like in the choline deficiency model [89], and there is a lack of poultry-specific research [93]. For example, the expression pattern of chicken DNA methyltransferase *DNMT3A* differs from that in mammals, suggesting that choline dosage may not be universally applicable [90].

### 5.2. DNA Methylation and Folliculogenesis

Avian follicular development is a complex and tightly regulated process that spans the entire reproductive cycle. It consists of two phases: prehierarchical and hierarchical follicles, both strictly controlled to guarantee optimal egg-laying performance [94]. With the onset of sexual maturity, the ovarian primordial follicle pool is activated, embarking on the growth phase and progressing to primary and mature follicles in preparation for ovulation [95]. Follicular development adheres to a strict hierarchy, gradually maturing from small to large [96]. Prehierarchical follicles are prone to atresia before entering the differentiation stage. In birds, hierarchical follicles usually consist of 3–6 follicles over a species-specific size threshold: chicken (>9 mm) [97], goose (>12 mm) [98], quail (>8 mm) [99].

In birds, from their formation to the early embryonic development after fertilization, germ cells experience significant epigenetic changes [100,101]. During epigenetic reprogramming, avian primordial germ cells only undergo limited CpG demethylation and retain numerous imprinted genes [101,102]. However, when developing into functional gametes, they face extensive epigenetic reprogramming [103]. The specific mechanisms regulating genomic methylation and remethylation in avian oocytes, particularly during follicle selection, warrant further investigation.

### 5.3. Choline Regulation of Steroid Hormone Synthesis and Follicular Development

Ovarian steroid hormones include progesterone, estrogens, and androgens. Current research indicates that estrogens, particularly estradiol, have the most significant impact on follicular development [104]. Neonatal mice were injected with 20 µg of E_2_ daily for 3 days starting at 1 day postpartum (1 dpp). This treatment significantly inhibited primordial follicle (PF) formation and reduced the number of PFs in ovaries at 4 dpp. E_2_ treatment also decreased cell proliferation, as indicated by the downregulation of Ki67 and Top2a. In 21 dpp ovaries, multiple oocyte follicles (MOFs) appeared, indicating disrupted PF formation. In vitro, ovarian cells from 4 dpp E_2_-treated mice were analyzed by single-cell RNA sequencing (scRNA-seq), revealing that E_2_ treatment disrupted the differentiation of pre-granulosa cells (PG) and altered mitochondrial activity in oocytes [105]. Estradiol is also a crucial factor for follicular development, differentiation, and survival, especially in the growth of antral follicles [106]. Estradiol increases the sensitivity of granulosa cells to FSH and LH, thereby promoting granulosa cell proliferation and follicular development [107]. Estrogens exert their effects after binding to their respective receptors. Two types of estrogen receptors, estrogen receptor alpha (*ERα*) and estrogen receptor beta (*ERβ*), have been identified in the ovary, with *ERβ* being expressed at significantly higher levels than *ERα* [108]. Knockout mice lacking *ERβ* exhibit infertility reduced follicular development, and lower ovulation rates [109,110,111], indicating that estradiol primarily promotes follicular development through the *ERβ* pathway. In a study investigating additions of 0, 200, 400, 600, 800, and 1000 mg/kg choline to the diet of laying geese, the content of E_2_ in serum and the number of small white follicles were significantly increased after adding 600 mg/kg choline, and the expression of the *ERα* gene in the ovaries was the highest in the 1000 mg/kg group, while the expression of *ERβ* was lowest in 1000 mg/kg group [68]. It can be seen from the above that appropriate choline supplementation affects the expression of *ERα* and *ERβ* through the hypothalamic–pituitary–gonadal axis, enhances the binding of E_2_ to its receptor, and promotes follicular development, thereby improving the reproductive performance of geese.

*Cyp19a1* encodes aromatase, which converts testosterone to E_2_. High temperatures cause hypermethylation of multiple CpG sites in the *Cyp19a1* promoter region of red-eared slider turtle embryos’ gonads. When this hypermethylation reduces aromatase activity, the E_2_/T ratio in the gonadal microenvironment becomes imbalanced. *ERα* can enhance *Cyp19a1* transcription by binding to the estrogen response element upstream of *Cyp19a1*, forming a positive feedback loop. This activates the SRY-Box transcription factor 9/doublesex and mab-3 related transcription factor 1 (*SOX9/DMRT1*) signaling pathway and induces testis differentiation [112,113,114,115]. In PCOS patients’ ovarian granulosa cells, *Cyp19a1* promoter methylation is significantly downregulated, and this downregulation is negatively correlated with gene expression [80]. *Cyp19a1* promoter methylation is negatively related to the E_2_/androstenedione ratio in follicular fluid, indicating it may suppress aromatase activity and increase androgen synthesis in granulosa cells [116]. Current research implies that choline could regulate *Cyp19a1* methylation by reducing the methylation level of CpG islands, removing the inhibition of *Cyp19a1* transcription, and restoring the function of aromatase, promoting androgen-to-estrogen conversion and alleviating PCOS-related hyperandrogenism. Mice aged 21–24 days were injected with equine chorionic gonadotropin to stimulate follicular growth, followed by human chorionic gonadotropin (hCG) injection 48 h later to induce ovulation and luteinization. Ovarian granulosa cells were collected at 0, 1, 2, 4, 8, and 12 h after hCG injection. The study found that hCG induction led to high H3K4me3 and low H3K9me3 and H3K27me3 at the Star promoter region. This increased C/EBPβ binding to the Star promoter, upregulating Star expression. For *Cyp19a1*, the promoter region saw low *H3K4me3* and high *H3K27me3*, reducing cyclic adenosine monophosphate response element binding and downregulating *Cyp19a1* expression [117]. Also, reducing anti-müllerian hormone receptor type 2 methylation may prevent excessive AMH-induced suppression of primordial follicle recruitment, thus preserving ovarian reserve [118,119,120]. Stimulating immortalized non-luteinized human granulosa cells (HGrC1) with bone morphogenetic protein 15 (*BMP15*) induces *FSHR* expression in human granulosa cells via Smad and non-Smad pathways and increases *Cyp19a1* mRNA expression and estradiol production [121]. Lowered *BMP15* levels may disrupt follicular development [122,123]. Choline can reduce follicle atresia and improve oocyte survival by blocking *BMP15*-mediated atresia and gene expression chaos from DNA hypomethylation [124,125]. Also, choline shortage can upset the competitive binding of *SF-1* and chicken ovalbumin upstream promoter transcription factor [126], possibly increasing the regulation of aromatase promoter I.4 by pro-inflammatory factors like interleukin-1 beta, thus promoting estrogen-dependent conditions such as endometriosis and uterine fibroids [127,128]. By discussing choline’s roles in these processes, this review provides theoretical foundations for future research and novel approaches to improve animal reproductive performance, contributing to the broader field of reproductive health [129,130]. To provide a clearer understanding, a schematic diagram of the proposed mechanisms of choline in follicular development is presented in Figure 1.

## 6. Conclusions

This review highlights the essential role of choline in ovarian follicular development. Adequate choline intake supports methylation balance, steroid hormone synthesis, and oocyte maturation, whereas deficiency or excess impairs follicular health through homocysteine accumulation or TMAO production. Overall, maintaining optimal choline levels is critical for ensuring a favorable follicular microenvironment and reproductive performance.

## Figures and Tables

**Figure 1 biology-14-01220-f001:**
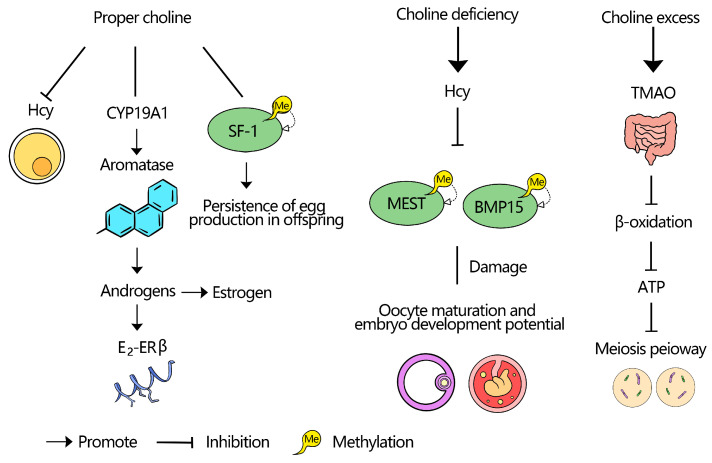
Proposed mechanisms of choline in follicular development. Choline participates in multiple pathways, including serving as a methyl donor through one-carbon metabolism, contributing to phosphatidylcholine synthesis for cell membrane integrity, and supporting acetylcholine production. These processes collectively influence DNA methylation, steroid hormone biosynthesis, and granulosa cell function, thereby regulating follicular growth and maturation.

**Table 1 biology-14-01220-t001:** Optimal dietary choline levels across species and their main reproductive or physiological effects.

Species	Optimal Choline Level	Main Effects
Japanese quail (Coturnix japonica)	0.126% (≈1260 mg/kg) diet; 1500 mg/kg to maintain laying performance and egg quality; 3500 mg/kg to enhance yolk antioxidant capacity	Improved egg weight and feed conversion; maintained egg quality; increased yolk antioxidant activity [66,67]
Laying geese	784–913 mg/kg diet	Enhanced reproductive performance, egg production, and sex hormone levels; promoted ovarian development and hormone synthesis [68]
Gestating sows	≈1910 mg/kg diet	Increased feed intake; improved piglet birth weight and litter uniformity; enhanced maternal antioxidant capacity and gut microbiota composition [69]
Pig	500–1000 mg/kg (diet)	Improves ovarian function, promotes corpus luteum formation, regulates reproduction-related gene expression, and enhances reproductive performance [70]
Transition Holstein cows	60 g/day rumen-protected choline (ReaShure^®^)	Reduced inflammatory gene expression in follicular cells; supported a healthier follicular environment during early postpartum [71]
Ewes (periconceptional and late gestation)	1.60 g/ewe/day rumen-protected choline	Reduced embryonic loss; increased lamb birth weight; improved maternal antioxidant status [72]
Lactating women	930 mg/day	Increased concentrations of PEMT-derived choline metabolites in breast milk; enhanced infant choline supply [73]
Pregnant women	550 mg/day	Increased maternal plasma choline metabolites; promoted fetal phospholipid synthesis via the PEMT pathway; supported maternal–fetal health [74]

## Data Availability

No new data were created or analyzed in this study. Data sharing is not applicable to this article.

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
