# Peer review of "Choline-Mediated Regulation of Follicular Growth: Interplay Between Steroid Synthesis, Epigenetics, and Oocyte Development"

_biology, 2025, doi:10.3390/biology14091220_

Round 1

Reviewer 1 Report

Comments and Suggestions for Authors

The authors have done a sufficient job in reviewing the role of choline in avian  reproductive function. However, this makes the title misleading for two reasons (see general comments below).

Likes: The authors have done a nice job describing how choline and choline metabolism occurs and can impact reproductive function.

Dislikes: The authors spend more time walking through the metobolic pathways and associated genes than actually connecting it to folliculogenesis.

General Comments:

The title suggests that the review will cover folliculogenesis but most of the sections rarely focus on folliculogenesis but discussion largely includes oogenesis and embryonic development.

The title also suggests that this review will cover most species and how choline impacts folliculogenesis but the information is largely avian based with rodent, human, and other species randomly sprinkled in.

Specific Comments:

Line 25: I think you intended this to be researchers or research but it is list as researces

Author Response

Comments 1:Likes: The authors have done a nice job describing how choline and choline metabolism occurs and can impact reproductive function.

Dislikes: The authors spend more time walking through the metobolic pathways and associated genes than actually connecting it to folliculogenesis.

Response 1: Thank you for your valuable feedback. I understand your concern regarding the balance between metabolic pathways and their connection to folliculogenesis. In response, I have added practical research on the effects of choline on follicular development in Section 4.3: Optimal Choline Levels for Follicular Health. This section now includes a detailed discussion on how choline influences actual follicular development, connecting the metabolic pathways with their impact on follicular health and maturation. I hope this addition better addresses the relationship between choline and folliculogenesis. Please let me know if any further revisions are needed. Page 4 and 5, 4.3. Optimal Choline Levels for Follicular Health, and line 185-219.

General Comments:

Comments 2: The title suggests that the review will cover folliculogenesis but most of the sections rarely focus on folliculogenesis but discussion largely includes oogenesis and embryonic development.

Response 2: Thank you for your valuable comment. Based on your feedback, we realized the importance of more clearly distinguishing between folliculogenesis and oogenesis in the title. In response, we have revised the title to explicitly include oocyte development. The new title, “Choline-Mediated Regulation of Follicular Growth: Interplay Between Steroid Synthesis, Epigenetics, and Oocyte Development,” not only focuses on the regulation of follicular growth but also highlights the close relationship between oocyte development and folliculogenesis. We hope this revision addresses your concern and more accurately reflects the content of the review. Page 1, Title, and line 2-4.

Comments 3: The title also suggests that this review will cover most species and how choline impacts folliculogenesis but the information is largely avian based with rodent, human, and other species randomly sprinkled in.

Response 3: Thank you for your valuable comment. In response to your suggestion, I have expanded the review to include more information on the effects of choline on both humans and other animal species. While much of the review was initially based on avian studies, I have now included additional data and discussions related to rodent, human, and other species to provide a more comprehensive overview of how choline impacts folliculogenesis across different species. I hope this revision addresses your concern and improves the balance of the review. Page 4 and 5, 4.3. Optimal Choline Levels for Follicular Health, and line 185-219.

Specific Comments:

Comments 4: Line 25: I think you intended this to be researchers or research but it is list as researces

Response 4: I have corrected "researces" to "researchers" as suggested. I appreciate your attention to detail. Page 1 , Abstract, and line 26.

Reviewer 2 Report

Comments and Suggestions for Authors

Reviewer Comments

This manuscript presents a well-conducted Choline Regulation of Follicular Development: Steroid Synthesis and Epigenetics. The study is robust, using large-scale GWAS data, and the analytical framework is appropriate. However, several areas require attention, including grammatical errors, improved clarity in writing, and alignment with existing literature to strengthen the review paper s scientific impact.

  • Tittle Suggestions
  1. "Choline’s Role in Follicular Development: Regulation of Steroidogenesis and Epigenetic Mechanisms”
  2. “Choline-Mediated Regulation of Follicular Growth: Interplay Between Steroid Synthesis and Epigenetics”
  3. “The Influence of Choline on Follicular Maturation: Steroid Production and Epigenetic Control”,
  • Suggestions and Comments:
  1. The in vitro culture systems, experimental conditions, and species-specific data need clearer description. Clarify whether this is a review or a meta-analysis incorporating original data.
  2. There are several grammatical and typographical errors throughout the manuscript. A professional language revision should be corrected.
  3. Include at least one diagram summarizing choline's effects on follicle development, hormone. synthesis, and epigenetic regulation. Previous study should be summarizing in table format.
  4. Introduction should be more better accoding to your study
  5. Lack of species specific data and limited experimental validation in poultry.
  6. No discussion on breed variability or long-term reproductive outcomes.
  7. The optimal choline dosage should be determined for different species in this review paper.
  8. Add a table summarizing species-wise choline effects on follicular development.
  9. All abbreviations should be done at first use.
  10. Consider moving technical content to supplementary materials.
  11. Line 25, Reseaces replace with research
  12. Line 35-37, Delete space before Steroidogenic factor, and rewrite with A phenomenon of steroidogenic factor 1 (SF-1) methylation has been observed in poultry, showing that choline affects offspring egg-laying persistence by altering the adrenal-ovarian axis DNA methylation imprint".
  13. Line 51-52, rephrase it with "gene expression regulation – indispensable biological processes during folliculogenesis".
  14. Line 67. Within replace with “while”.
  15. Line 80-81, replace with "only one-third of dietary choline is absorbed intact, while the remaining two-thirds undergo microbial metabolism".
  16. Line 125 replace with "mechanisms, which lead to an overall suppression of choline incorporation into the CDP-choline pathway".
  17. Line 161, corrected with "associated with poor reproductive outcomes."
  18. Line 171, corrected with "Choline supplementation can restore methylation patterns, thereby stabilizing metabolism and endocrine function."

Citation:

Recommended Citations to Strengthen for this Study. Add in Introduction

  1. Qian, F., Zhou, L., Li, Y., Yu, Z., Li, L., Wang, Y.,... Li, C. (2023). SEanalysis 2.0: a comprehensive super-enhancer regulatory network analysis tool for human and mouse. Nucleic Acids Research, 51(W1), W520-W527. doi: https://doi.org/10.1093/nar/gkad408
  2. Gao, Y., Wang, C., Wang, K., He, C., Hu, K.,... Liang, M. (2022). The effects and molecular mechanism of heat stress on spermatogenesis and the mitigation measures. Systems Biology in Reproductive Medicine, 68(5-6), 331-347. doi: https://doi.org/10.1080/19396368.2022.2074325
  3. Zhu, Q., Sun, J., An, C., Li, X., Xu, S., He, Y.,... Liang, M. (2024). Mechanism of LncRNA Gm2044 in germ cell development. Frontiers in Cell and Developmental Biology, 12, 1410914. doi: https://doi.org/10.3389/fcell.2024.1410914
  4. Chang, H., Wang, D., Xia, W., Pan, X., Huo, W., Xu, S.,... Li, Y. (2016). Epigenetic disruption and glucose homeostasis changes following low-dose maternal bisphenol A exposure. Toxicology Research, 5(5), 1400-1409. doi: 10.1039/c6tx00047a
  5. Zhou, J., Guo, Z., Peng, X., Wu, B., Meng, Q., Lu, X.,... Guo, T. (2025). Chrysotoxine regulates ferroptosis and the PI3K/AKT/mTOR pathway to prevent cervical cancer. Journal of Ethnopharmacology, 338, 119126. doi: https://doi.org/10.1016/j.jep.2024.119126
  6. Zhang, G., Song, C., Yin, M., Liu, L., Zhang, Y., Li, Y.,... Li, C. (2025). TRAPT: a multi-stage fused deep learning framework for predicting transcriptional regulators based on large-scale epigenomic data. Nature Communications, 16(1), 3611. doi: 10.1038/s41467-025-58921-0

Author Response

Comments 1: "Choline’s Role in Follicular Development: Regulation of Steroidogenesis and Epigenetic Mechanisms”; “Choline-Mediated Regulation of Follicular Growth: Interplay Between Steroid Synthesis and Epigenetics”; “The Influence of Choline on Follicular Maturation: Steroid Production and Epigenetic Control”.

Response 1: Thank you for your helpful suggestion. In response to your feedback, we have modified the title to better reflect the importance of species diversity, ensuring that the title encompasses the relevance across different species. Additionally, we have included oocyte development in the title to emphasize how choline-mediated regulation of follicular growth is related to the process of oocyte maturation. The revised title, “Choline-Mediated Regulation of Follicular Growth: Interplay Between Steroid Synthesis, Epigenetics, and Oocyte Development,” highlights both the role of choline in follicular growth and its influence on oocyte development while addressing the issue of species diversity. We hope this revision satisfies your comment and better aligns with the scope of the review. Page 1, Title, and line 2-4.

Suggestions and Comments:

Comments 2: The in vitro culture systems, experimental conditions, and species-specific data need clearer description. Clarify whether this is a review or a meta-analysis incorporating original data.

Response 2: Thank you for your valuable feedback. This manuscript is a review, not a meta-analysis, and it synthesizes findings from existing studies along with some original data from our laboratory. For species-specific data, we included studies involving various species, highlighting how different species respond to choline supplementation in terms of follicular development. Specific experimental conditions, dosages, and outcomes have been clarified in the updated manuscript. Page 8, 5.3. Choline Regulation of Steroid Hormone Synthesis and Follicular Development, and line 281-288.

Comments 3: There are several grammatical and typographical errors throughout the manuscript. A professional language revision should be corrected.

Response 3: Thank you for your valuable feedback. I appreciate you pointing out the grammatical and typographical errors in the manuscript. I have thoroughly reviewed the manuscript and made the necessary corrections to ensure proper grammar and formatting. Page 1, Abstract, and line 25; Page 3, 3. Choline Structure, Absorption and Metabolism, and line 103; Page 3, 3. Choline Structure, Absorption and Metabolism, and line 116; Page 4, 4.2. Toxicity, and line 182.

Comments 4: Include at least one diagram summarizing choline's effects on follicle development, hormone. synthesis, and epigenetic regulation. Previous study should be summarizing in table format.

Response 4 :Thank you for your valuable suggestion. I have addressed your request as follows:

The mechanism of choline’s effects on follicular development, hormone synthesis, and epigenetic regulation has been summarized in the Graphical Abstract, which is now included in the manuscript. This diagram provides a clear visual summary of the pathways involved and how choline influences these biological processes.

The previous studies have been summarized in Table 1, where key research findings and experimental methods are clearly presented for easy comparison. Page 4 and 5, 4.3. Optimal Choline Levels for Follicular Health, and line 185-219; Graphical Abstract.

Comments 5: Introduction should be more better accoding to your study

Response 5: Thank you for your helpful comment. I have revised the Introduction section to better align with the objectives of the study, as per your suggestion. Additionally, I have incorporated the references you recommended to further strengthen the context and background of the manuscript. Page 2, 1. Introduction, and line 51-56; 59-68; 71-74.

Comments 6: Lack of species specific data and limited experimental validation in poultry.

Response 6: Thank you for your valuable comment. In response to your concern regarding the lack of species-specific data and limited experimental validation in poultry, I have added relevant information in Section 4.3: Optimal Choline Levels for Follicular Health. This section now includes more species-specific data and experimental findings related to poultry, addressing the gap you mentioned.

Additionally, I have incorporated a table summarizing the key data and experimental results to further clarify the species-specific effects of choline on follicular health in poultry. Page 4 and 5, 4.3. Optimal Choline Levels for Follicular Health, and line 185-219.

Comments 7: No discussion on breed variability or long-term reproductive outcomes.

Response 7: Thank you for your valuable comment. In response to your concern regarding the lack of discussion on breed variability and long-term reproductive outcomes, I have added relevant information in Section 4.3: Optimal Choline Levels for Follicular Health. This section now includes a discussion on the breed-specific variability in choline requirements and its impact on long-term reproductive outcomes in different species, including poultry. Page 4 and 5, 4.3. Optimal Choline Levels for Follicular Health, and line 185-219.

Comments 8: The optimal choline dosage should be determined for different species in this review paper. Add a table summarizing species-wise choline effects on follicular development.

Response 8: Thank you for your helpful comment. In response to your suggestion regarding the determination of the optimal choline dosage for different species, I have added relevant information in Section 4.3: Optimal Choline Levels for Follicular Health. This section now includes a discussion on the optimal choline levels for different species, highlighting species-specific differences in choline requirements and their effects on follicular development. Page 4 and 5, 4.3. Optimal Choline Levels for Follicular Health, and line 185-219

Comments 9: All abbreviations should be done at first use.

Response 9: Thank you for your helpful comment. I have thoroughly reviewed the manuscript and ensured that all abbreviations are spelled out at first use. Any abbreviations that were previously not defined have now been properly introduced and explained. Page 2 , 1.Introduction, and line 65; Page 4 , 4.1. Deficiency, and line 158-159, 161; Page 4 , 4.2. Toxicity, and line 172; Page 6 , 5. Mechanism of Choline in Follicular Development, and line 223-224; Page 7 , 5.1. Choline and DNA Methylation, and line 241; Page 8 , 5.3. Choline Regulation of Steroid Hormone Synthesis and Follicular Development, and line 293, 302; 310, 328, 331.

Comments 10: Consider moving technical content to supplementary materials.

Response 10: Thank you for your valuable feedback. In response to your suggestion, I have moved the more technical content, including detailed metabolic pathways, enzyme activities, and specific experimental methods, to the Supplementary Materials section. This has allowed me to streamline the main text and focus on the key findings and discussions. Additionally, I have made necessary deletions and modifications to ensure that the content remains clear and concise while maintaining scientific accuracy. I hope these revisions address your concerns. Please let me know if any further changes are needed. Page 3 , 3. Choline Structure, Absorption and Metabolism, and line 104-105, 123-125, 136-138; Page 4 , 4.1. Deficiency, and line 146; Page 6 , 5. Mechanism of Choline in Follicular Development, and line 221,228-229; Page 6, 5.2. DNA Methylation and Folliculogenesis, and line 271.

Supplementary Materials

Choline metabolism involves multiple complex biochemical pathways. As a component of all cellular membrane phospholipids, all tissues can accumulate choline via facilitated diffusion or transporter assistance. The liver, kidney, mammary gland, placenta, and brain show particularly high accumulation.

Through microbial action, CHDH oxidizes choline into TMA, subsequently oxidized by hepatic flavin-containing monooxygenases (FMOs) into trimethylamine N-oxide  (TMAO). TMAO is associated with increased cardiovascular risk.

Choline can alternatively react with CDP-diacylglycerol, catalyzed by choline phosphate cytidylyltransferase, to generate phosphatidylethanolamine (PE) PE is then converted into PC via Phosphatidylethanolamine N-methyltransferase (PEMT)-catalyzed N-methylation, using S-adenosylmethionine (SAM)-derived methyl groups with vitamin B12 as cofactor. During choline deficiency, choline is preferentially directed toward PC synthesis to meet membrane requirements.

A portion of choline enters the mitochondria via CTL1 transporters on the inner membrane. It is converted into betaine aldehyde by choline dehydrogenase (CHDH), then into betaine by betaine aldehyde dehydrogenase, thereby providing methyl groups.

In the intestinal lumen, PC hydrolyzes into choline. Both pancreatic juice and mucosal cells contain PC-hydrolyzing enzymes. Phospholipase A2, present in pancreatic juice and intestinal brush border cleaves β-fatty acid molecules. While intestinal mucosal cells, phospholipase A1 cleaves α-fatty acids while phospholipase B processes both α- and β-fatty acids.

Choline, via its metabolite betaine, participates in HCY metabolism, impacts one-carbon metabolism, and ultimately affects SAM synthesis, thereby influencing cellular methylation. SAM, a key methyl donor, can influence HCY methylation and methionine regeneration. Its deficiency may lead to HCY accumulation and increased plasma HCY levels [48,49]. Thus, choline deficiency may increase HCY levels, which can negatively affect oocyte and embryo quality and developmental dynamics.

Choline is oxidized into betaine, which, via betaine-homocysteine methyltransferase (BHMT)-enzyme-catalyzed remethylation of HCY to MET, directly determines SAM biosynthesis efficiency.

These follicles are arranged in a size hierarchy (F1-F5 or more), with F1 being the next to ovulate. After F1 ovulates, a new large yellow follicle enters the hierarchical pool, a process called follicle selection. Small white and large white follicles continuously supply new follicles to the pool. Follicle selection is a key step in determining bird egg-laying output.

Comments 11: Line 25, Reseaces replace with research

Response 11: The word “reseaces” has been replaced with “research”. I appreciate you pointing this out. I have corrected the typo accordingly. Page 1, Abstract, and line 26.

Comments 12: Line 35-37, Delete space before Steroidogenic factor, and rewrite with A phenomenon of steroidogenic factor 1 (SF-1) methylation has been observed in poultry, showing that choline affects offspring egg-laying persistence by altering the adrenal-ovarian axis DNA methylation imprint".

Response 12: I have deleted the space before “Steroidogenic factor” and rephrased the sentence to:

“A phenomenon of steroidogenic factor 1 (SF-1) methylation has been observed in poultry, showing that choline affects offspring egg-laying persistence by altering the adrenal-ovarian axis DNA methylation imprint.” Page 1, Abstract, and line 36-38.

Comments 13: Line 51-52, rephrase it with "gene expression regulation – indispensable biological processes during folliculogenesis".

Response 13: I have rephrased the sentence as: ”Moreover, choline participates in DNA methylation and the regulation of gene expression, both of which are essential biological processes during folliculogenesis.”  Page 2, 1. Introduction, and line 51-53.

Comments 14: Line 67. Within replace with “while”.

Response 14: Thank you for your comment regarding Line 67. In response to your suggestion, I initially replaced "within" with "while," but after considering other reviewer comments, the entire sentence was revised and removed. As a result, the word "while" was also deleted as part of these changes.

I hope this clarifies the situation. Please let me know if any further revisions are needed.

Comments 15: Line 80-81, replace with "only one-third of dietary choline is absorbed intact, while the remaining two-thirds undergo microbial metabolism".

Response 15: The sentence has been revised to:

“Only one-third of dietary choline is absorbed intact, while the remaining two-thirds undergo microbial metabolism.” Page 3, 3. Choline Structure, Absorption and Metabolism, and line 116-117.

Comments 16: Line 125 replace with "mechanisms, which lead to an overall suppression of choline incorporation into the CDP-choline pathway".

Response 16: I have replaced the sentence with:

“mechanisms, which lead to an overall suppression of choline incorporation into the CDP-choline pathway.” Page 4, 4.1. Deficiency, and line 146-147.

Comments 17: Line 161, corrected with "associated with poor reproductive outcomes."

Response 17: The sentence has been corrected to: “associated with poor reproductive outcomes.” I appreciate your attention to detail. I have corrected this accordingly. Page 4, 4.2. Toxicity, and line 182.

Comments 18: Line 171, corrected with "Choline supplementation can restore methylation patterns, thereby stabilizing metabolism and endocrine function."

Response 18: I have corrected the sentence to: “Choline supplementation can restore methylation patterns, thereby stabilizing metabolism and endocrine function.” Page 6, 5. Mechanism of Choline in Follicular Development, and line 228-229.

Comments 19: Recommended Citations to Strengthen for this Study. Add in Introduction

Response 19: Thank you for your helpful suggestion regarding the recommended citations. I have added the relevant references to the Introduction section to strengthen the study, as you recommended. The updated references now reflect the current understanding and recent findings in the field, ensuring that the manuscript is well-supported by recent research. I appreciate your input and hope the updated version meets your expectations. Page 2, 1. Introduction, and line 51-56, 59-68, 71-74.

Citation:

Qian, F., Zhou, L., Li, Y., Yu, Z., Li, L., Wang, Y.,... Li, C. (2023). SEanalysis 2.0: a comprehensive super-enhancer regulatory network analysis tool for human and mouse. Nucleic Acids Research, 51(W1), W520-W527. doi: https://doi.org/10.1093/nar/gkad408IF: 13.1 Q1

Gao, Y., Wang, C., Wang, K., He, C., Hu, K.,... Liang, M. (2022). The effects and molecular mechanism of heat stress on spermatogenesis and the mitigation measures. Systems Biology in Reproductive Medicine, 68(5-6), 331-347. doi: https://doi.org/10.1080/19396368.2022.2074325IF: 2.2 Q2

Zhu, Q., Sun, J., An, C., Li, X., Xu, S., He, Y.,... Liang, M. (2024). Mechanism of LncRNA Gm2044 in germ cell development. Frontiers in Cell and Developmental Biology, 12, 1410914. doi: https://doi.org/10.3389/fcell.2024.1410914IF: 4.3 Q1

Chang, H., Wang, D., Xia, W., Pan, X., Huo, W., Xu, S.,... Li, Y. (2016). Epigenetic disruption and glucose homeostasis changes following low-dose maternal bisphenol A exposure. Toxicology Research, 5(5), 1400-1409. doi: 10.1039/c6tx00047aIF: 2.1 Q3

Zhou, J., Guo, Z., Peng, X., Wu, B., Meng, Q., Lu, X.,... Guo, T. (2025). Chrysotoxine regulates ferroptosis and the PI3K/AKT/mTOR pathway to prevent cervical cancer. Journal of Ethnopharmacology, 338, 119126. doi: https://doi.org/10.1016/j.jep.2024.119126IF: 5.4 Q1

Zhang, G., Song, C., Yin, M., Liu, L., Zhang, Y., Li, Y.,... Li, C. (2025). TRAPT: a multi-stage fused deep learning framework for predicting transcriptional regulators based on large-scale epigenomic data. Nature Communications, 16(1), 3611. doi: 10.1038/s41467-025-58921-0IF: 15.7 Q1.

Reviewer 3 Report

Comments and Suggestions for Authors

The effects of choline on steroidogenesis have been extensively studied, with over 30 review articles published in recent decades. While steroids such as estrogen and androgen can influence follicular development, limited attention has been paid on the regulatory effect of choline on follicular development. Therefore, I think this mini-review is acceptable for publication, but the manuscript cannot be published as is.

The following issues need addressed:

This is a mini-review and there is no methodology in the manuscript. I strongly suggest the authors write a brief method and explain the literature search time, database, and how the literature were reviewed, what kind of key words were used.....

As this review focuses on ovarian follicular development, what about you just use the title: "The regulation of choline on ovarian follicular development" or you choose a similar one. The Steroid Synthesis and Epigenetics are partial mechanism of how choline regulate follicular development. 

If it is possible, please clearly address the effect of choline on humans and animals.

Please clearly address this is a review article in the summary and abstract. Most readers will mistakenly assume this is a research article if they only read the summary. 

Author Response

Comments 1: This is a mini-review and there is no methodology in the manuscript. I strongly suggest the authors write a brief method and explain the literature search time, database, and how the literature were reviewed, what kind of key words were used.....

Response 1: Thank you for your valuable comment and suggestion. In response to your feedback, I have added a section titled "2. Literature Review Methodology" to the manuscript. This section now includes a brief explanation of the literature search process, specifying the time frame, databases used, and the keywords employed in the search. Additionally, I have outlined the criteria for how the literature was reviewed and selected. Page 2, 2. Literature Review Methodology, and line 75-97.

Comments 2: As this review focuses on ovarian follicular development, what about you just use the title: "The regulation of choline on ovarian follicular development" or you choose a similar one. The Steroid Synthesis and Epigenetics are partial mechanism of how choline regulate follicular development.

Response 2: Thank you for your valuable suggestion regarding the title. After carefully considering your feedback and other reviewers' comments, I have updated the title to: "Choline-Mediated Regulation of Follicular Growth: Interplay Between Steroid Synthesis, Epigenetics, and Oocyte Development."

I believe this new title better reflects the scope of the review, which focuses on ovarian follicular development, while also addressing the critical mechanisms through steroid synthesis, epigenetics, and oocyte development. These factors play an integral role in how choline regulates follicular growth, and I hope this title more accurately represents the review's content. Page 1, Title, and line 2-4.

Comments 3: If it is possible, please clearly address the effect of choline on humans and animals.

Response 3: Thank you for your valuable suggestion. In response to your comment regarding the effect of choline on humans and animals, I have added detailed information in Section 4.3. Optimal Choline Levels for Follicular Health. This section now includes a discussion on the specific effects of choline on humans and various animal species, including their impact on follicular development, reproductive health, and overall metabolic function. Page 4 and 5, 4.3. Optimal Choline Levels for Follicular Health, and line 185-219.

Comments 4: Please clearly address this is a review article in the summary and abstract. Most readers will mistakenly assume this is a research article if they only read the summary.

Response 3: Thank you for your valuable comment. I have made the necessary changes in the summary and abstract to clearly indicate that this is a review article. I have added specific wording to ensure that readers understand the nature of the manuscript, preventing any confusion with a research article.

I hope these revisions address your concern. Please let me know if any further changes are needed. Page 1, Simply Summary, and line 11; Page 1, Abstract, and line 23-24; Page 9, 6.Conclusion, and line 342.

Round 2

Reviewer 2 Report

Comments and Suggestions for Authors

Reviewer Comments

This manuscript presents a well-conducted Choline Regulation of Follicular Development: Steroid Synthesis and Epigenetics. The study is robust, using large-scale GWAS data, and the analytical framework is appropriate. However, several areas require attention, including grammatical errors, improved clarity in writing, and alignment with existing literature to strengthen the review paper s scientific impact.

  • Suggestions and Comments:

All Comments should be completed and add in review paper

  1. Line 51, Add Reference No. 11 with statement after reference No. 4,5, remove from line 65 to 68.
  2. Line 340, Add Reference No. 12,13 with statement after reference No. 128,129, remove from line 71 to 74.
  3. In Table 1, plz add column for References according to previous study.
  4. Choline mechanism should be added in diagram form.
  5. Conclusion should be proper and short according to your review objective.

Author Response

Comments 1: Line 51, Add Reference No. 11 with statement after reference No. 4,5, remove from line 65 to 68.

Response 1: Thank you for the suggestion. I have revised the manuscript accordingly. Reference No. 11 has been added after references No. 4 and 5 at line 51, and the section from line 65 to 68 has been removed as requested. Page 2, 1. Introduction, and line 51-55.

Comments 2: Line 340, Add Reference No. 12,13 with statement after reference No. 128,129, remove from line 71 to 74.

Response 2: Thank you for the comment. I have revised the manuscript as suggested. References No. 12 and 13 have been added after references No. 128 and 129 at line 340, and the section from line 71 to 74 has been removed. Page 9, 5.3. Choline Regulation of Steroid Hormone Synthesis and Follicular Development, and line 341-343.

Comments 3: In Table 1, plz add column for References according to previous study.

Response 3: Thank you for the suggestion. I have revised Table 1 by adding a column for References according to the previous studies. Page 5 and 6, 4.3. Optimal Choline Levels for Follicular Health, Table 1, and line 219.

Comments 4: Choline mechanism should be added in diagram form.

Response 4: Thank you for the valuable suggestion. A diagram illustrating the mechanism of choline has been added to the revised manuscript to provide a clearer understanding. Page 9, 5.3. Choline Regulation of Steroid Hormone Synthesis and Follicular Development, and line 344-352.

Comments 5: Conclusion should be proper and short according to your review objective.

Response 5: Thank you for the comment. The conclusion has been revised to be more concise and aligned with the main objectives of this review. Page 9, 6. Conclusion, and line 354-359.

Reviewer 3 Report

Comments and Suggestions for Authors

The revised version is much better, I think it is ready to be published.

Author Response

Comments 1: The revised version is much better, I think it is ready to be published.

Response 1: Thank you very much for your positive feedback. I truly appreciate your time and effort in reviewing our manuscript.
